# Abyssomicins—A 20-Year Retrospective View

**DOI:** 10.3390/md19060299

**Published:** 2021-05-24

**Authors:** Hans-Peter Fiedler

**Affiliations:** Department of Microbiology/Biotechnology, Interfaculty Institute of Microbiology and Infection Medicine Tübingen (IMIT), University of Tuebingen, Auf der Morgenstelle 28, D-72076 Tübingen, Germany; hans-peter.fiedler@uni-tuebingen.de

**Keywords:** abyssomicins, chorismate pathway, 4-amino-4-deoxychorismate synthase inhibitor

## Abstract

Abyssomicins represent a new family of polycyclic macrolactones. The first described compounds of the abyssomicin family were abyssomicin B, C, atrop-C, and D, produced by the marine actinomycete strain *Verrucosispora maris* AB-18-032, which was isolated from a sediment collected in the Sea of Japan. Among the described abyssomicins, only abyssomicin C and atrop-abyssomicin C show a high antibiotic activity against Gram-positive bacteria, including multi-resistant and vancomycin-resistant strains. The inhibitory activity is caused by a selective inhibition of the enzyme 4-amino-4-deoxychorismate synthase, which catalyzes the transformation of chorismate to para-aminobenzoic acid, an intermediate in the folic acid pathway.

## 1. Introduction

An efficient search for new bioactive compounds, such as antibiotics, demands specific requirements: 

First, there is a need for a massive set of taxonomically characterized and dereplicated microorganisms isolated from terrestrial and/or marine habitats, with a high potency of producing unique secondary metabolites. A representative example of such a source are members of the order Actinomycetales, which are known as potent producers of unique secondary metabolites. Second, there needs to be a specific target that is essential for a pathogenic organism but is not present in humans in order to provide selectivity in toxic effects. Third, highly efficient analytic equipment and techniques are required to identify novel secondary metabolites in the culture broth or extracts from mycelium and culture filtrate, such as the combination of HPLC with diode array detection and/or mass spectrometry and characterization employing a home-made or commercial database. One of our screening programs was focused on the inhibition of the chorismate pathway, taking over an idea of Emeritus Professor Hans Zähner, one of the pioneers in antibiotic research and chair of the Institute of Microbiology for over 30 years at the University of Tübingen, who retired in 1994. The chorismate pathway leads to the biosynthesis of the aromatic amino acids, and to the biosynthesis of *para*-aminobenzoic acid (*p*Aba), an intermediate in the biosynthesis of folic acid. This pathway is present in plants, fungi, prokaryotes, and some parasites, such as *Plasmodium* and *Toxoplasma*, but it is not present in humans (Figure 1). The biosynthesis of *p*Aba is catalyzed by two enzymes, 4-amino-4-deoxychorismic acid (ADC) synthase and ADC lyase, which were the target of our screening concept. 

As promising candidates for the production of new secondary metabolites, we selected 201 characterized strains from our own collection and strains from my collaborators Alan T. Bull (University of Kent) and Michael Goodfellow (Newcastle University) belonging to the families *Streptomycetaceae* and *Micromonosporaceae,* and rare actinomycete genera, isolated both from terrestrial and marine habitats. Extracts were subjected to the screening assay based on a simple but highly selective agar plate diffusion assay. Only one extract obtained from strain AB-18-032, a marine isolate from a sediment collected from the Sea of Japan belonging to the family *Micromonosporaceae*, inhibited *p*Aba biosynthesis selectively. The secondary metabolite pattern of the active extract was analyzed by HPLC-diode array, whereby three metabolites were characterized by our HPLC-UV-Vis Database as presumably new compounds, which we named abyssomicins.

The first announcement of abyssomicins was done two years later in 2003 in a patent application [1] because of their pharmacologically relevant properties which favored their application as promising therapeutics against infection diseases, especially of pathogenic multi-resistant Gram-positive bacteria, including vancomycin-resistant *Staphylococcus aureus*. Subsequently, we published the screening method, fermentation, isolation, and biological activities of abyssomicins and the taxonomy of the producing strain [2]. Structure elucidation of abyssomicins and their mode of action was done by the group of Roderich Süssmuth (Technical University Berlin) [3,4,5]. The complete characterization of strain AB-18-032 applying polyphasic taxonomy was done by the group of Michael Goodfellow (Newcastle University). Strain AB-18-032 was classified in the genus *Verrucosispora* as a new species and was named *Verrucosispora maris* sp. nov. as a type strain [6]. Investigations in the genetic analysis and description of the whole genome sequence of *Verrucosispora maris* AB-18-023 were done by the groups of Roderich Süssmuth and James E.M Stach (Newcastle University) [7,8]. The structures of the main compounds produced by *Verrucosispora maris* AB-18-023, abyssomicin C and atrop-abyssomicin C, are shown in Figure 2.

During the last years, numerous members of the abyssomicin family were isolated from actinomycetes from marine and terrestrial habitats, which was summarized in two excellent review articles [9,10]. Compared with all other described members of the abyssomicin family, only abyssomicin C and atrop-abyssomicin C, the main products of *Verrucosispora maris* AB18-032, showed potent inhibitory activity against Gram-positive bacteria, caused by the specific inhibition of the *p*-aminobenzoic acid/tetrahydrofolate biosynthetic pathway.

## 2. The Screening Concept

As described above, we have been choosing the chorismate pathway and its link to the *p*Aba biosynthesis as the target for our search of new inhibitors, because at this time, no specific natural inhibitor of *p*Aba biosynthesis was known that results in an inhibition of the folic acid biosynthesis, which is essential for microorganisms but not for humans [2].

In a pre-screen, we tested extracts of actinomycetes in a filter disk agar plate diffusion assay with *Bacillus subtilis* as the test organism. The only extracts of interest were those showing an inhibition zone grown on a chemically defined medium, and no inhibition zone grown on a complex medium. That indicates an antagonistic effect caused by a constituent of the complex medium. These extracts were selected for an antagonism assay that enabled us to distinguish between an inhibition of the aromatic amino acids pathway Tyr/Phe/Trp and the pathway of *p*Aba, and simultaneously the inhibition of the pathway prior to chorismate. Four agar plates were prepared seeded with *B. subtilis* in a chemically defined medium. A filter paper strip was soaked with an extract and placed on the agar plate. Across the antibiotic containing strip, a second strip was placed that was soaked with a solution of (a) Tyr+Trp+Phe+*p*Aba, (b) Tyr+Phe, (c) Trp, or (d) *p*Aba. When the growth inhibition is competitively reversed exclusively by (a), an inhibitor of an enzyme in front of chorismate can be assumed. An inhibitor of the biosynthetic pathway from chorismate to *p*Aba can be expected when the growth inhibition caused by an extract is competitively reversed by (d). The observance of inhibition zones for (b) and (c) indicates the detection of inhibitors of the aromatic amino acid pathways, Tyr/Phe and Trp, respectively.

This screening program included 201 actinomycetes strains from our own strain collection and strains that we obtained from our collaborators Alan Bull (University of Kent) and Michael Goodfellow (Newcastle University). These were in detail 104 terrestrial members of the family *Streptomycetaceae*, 33 members of the family *Micromonosporaceae* (8 terrestrial, 25 marine), and 64 members of rare actinomycete taxa (55 terrestrial, 9 marine). The actinomycetes strains were cultivated in various media, and extracts of their culture filtrates and mycelia were prepared, resulting in 930 extracts that were applied to the assay.

Among all extracts, only one extract from marine strain AB-18-032 inhibited selectively the pathway from chorismate to *p*Aba, indicating an inhibitor of the *p*Aba biosynthesis. The secondary metabolite profile of the active extract determined by HPLC-diode array analysis showed three peaks that could not be assigned to any known antibiotic by means of our HPLC-UV-Vis Database (Figure 3).

## 3. Fermentation and Isolation

Batch fermentations of strain AB-18-032 were carried out in the 10-litre scale in a complex medium and yielded 60 mg/L of abyssomicin C and atrop-C after a fermentation time of 96 h. The fermentation broth was separated by multiple sheet filtration into culture filtrate and mycelium. Abyssomicins were isolated from the culture filtrate by Amberlite XAD-16 chromatography, desorbed with a step gradient H_2_O-MeOH, and extracted after concentration with ethyl acetate at pH 5. Abyssomicins were purified by subsequent column chromatography on Sephadex LH-20 using MeOH as eluent, and on silica-diol with CH_2_Cl_2_-MeOH gradient elution [2]. Pure abyssomicin C and atrop-C, respectively, were obtained by preparative reversed-phase HPLC on Nucleosil-C18 and H_2_O-MeOH gradient elution, and lyophilization of the separated eluates.

## 4. Structure Elucidation and Synthesis Strategies

From abyssomicin B, C, and D, single crystals were obtained, and their relative configuration was determined by X-ray structure. NMR analyses confirmed the X-ray data [3]. The absolute stereochemistry was determined by both the Mosher method [11] and the Helmchen method [12].

A detailed analysis of the culture filtrate from fermentations of strain AB-18-032 revealed two additional signals in the chromatogram of HPLC-DAD-ESI-MS runs that were related to abyssomicins according to their UV-visible properties, and besides abyssomicin C, the presence of atrop-abyssomicin C as the main product of strain AB-18-032 could be confirmed [4]. The compounds, abyssomicin G and H, were isolated from the culture filtrate of the fermentation of strain AB-18-032 in analogy to abyssomicins B–D, and the purified compounds were analyzed by mass spectrometry and 1D and 2D NMR spectroscopy. The structures of all abyssomicins isolated from strain AB-18-032 are shown in Figure 4.

Abyssomicin C was the topic of several total synthetic strategies, based on its unique and complex spirotetronate structure, and on the need for structure optimization with regard to clinical applications. The group of Erik J. Sorensen published a Diels–Alder macrocyclization that enabled an efficient asymmetric synthesis of abyssomicin C [13]. The group of Martin E. Maier prepared the synthesis of the core structure of abyssomicin C containing an oxybicyclooctane ring and a tetronate by a Diels–Alder strategy [14]. A further strategy for the synthesis of abyssomicins C and D was reported by the group of Barry B. Snider [15]. While working on another route for the total synthesis of abyssomicin C, the group of Kyriacos C. Nicolaou discovered atrop-abyssomicin C, a novel isomer of abyssomicin C [16]. Atrop-abyssomicin C was found simultaneously as a natural product of *Verrucosispora maris* AB-18-032 [4]. The importance of spirotetronate antibiotics and their improved synthetic routes are highlighted in two review articles by the group of E.A. Theodorakis [17,18]. 

Continuous interest in abyssomicins has been shown by Vidali et al., applying biomimetic approaches toward the synthesis of abyssomicin C and atrop-abyssomicin C based on an intramolecular Diels–Alder reaction of a butenolide derivative attached to a keto-triene side chain [19].

## 5. Biological Activity and Mode of Action

Among all abyssomicins, only abyssomicin C and atrop-abyssomicin C showed a strong antimicrobial activity, which is restricted to Gram-positive bacteria. The MIC value of atrop-abyssomicin C against multi-resistant *Staphylococcus aureus* N313 was 3.5 µg/mL. MIC values in the range of 13–16 µg/mL were determined against vancomycin-resistant *S. aureus* Mu50, *Enterococcus faecalis* VanA, VanB, and *E. faecium* VanA, and against multi-resistant *S. epidermidis* CNS 184. Abyssomicin C showed 1.5-fold lowered MIC values compared to atrop-abyssomicin C. Gram-negative bacteria, filamentous fungi, and yeasts were not sensitive against abyssomicin C and atrop-abyssomicin C [2,4].

The antitumor activity of atrop-abyssomicin C was tested against various human tumor cell lines, such as HM02 (gastric adenocarcinoma), Hep G2 (hepatocellular carcinoma), and MCF7 (breast carcinoma), exhibiting a moderate inhibition with GI_50_ values of 7.9, 1.5, and 6.1 µg/mL, respectively. Atrop-abyssomicin C showed a slight activity against *Trypanosoma brucei rhodesiense*, with an IC_50_ value of 0.68 µg/mL.

Abyssomicin C and atrop-abyssomicin C were the first known natural inhibitors of *p*Aba biosynthesis. Two enzymes catalyze *p*Aba biosynthesis from chorismate, 4-amino-4-deoxychorismate (ADC) synthase, which converts chorismate and glutamine into ADC and glutamate, and ADC lyase, which catalyzes an elimination reaction of ADC to produce *p*Aba. ADC synthase is a heterodimer composed of two nonidentical subunits, PabA and PabB. PabA functions as a glutamine amidotransferase, while PabB catalyzes the substitution of the chorismate 4-hydroxy group by an amino group while retaining the original configuration. The Süssmuth group has shown that the PabB subunit of ADC synthase is the molecular target of abyssomicin C and atrop-abyssomicin C. The antibiotics act as covalent binders in a Michael addition to the side chain of Cys 263, located in the proximity of the active site of PabB, and inhibit *p*Aba formation and, consequently, folate biosynthesis [5]. The increased antimicrobial activity of atrop-abyssomicin C can be explained by its more powerful Michael acceptor properties [16].

The oxabicyclooctane ring system of abyssomicin C, and atrop-abyssomicin B shows a striking similarity to one solution conformation of chorismate, which suggests that both antibiotics act as substrate mimetics (Figure 5).

## 6. Taxonomy of the Producing Strain

Strain AB-18-032 was isolated from a sediment collected from the Sea of Japan at a depth of 289 meters in August 1991 as an outcome of the collaboration of Alan T. Bull (University of Kent) and Koki Horikoshi (JAMSTEC Tokyo). The strain was assigned to the family *Micromonosporaceae* based on its morphological and chemotaxonomic properties [2]. A scanning electron micrograph is shown in Figure 6.

The comparison of the nearly complete 16S rDNA gene sequence of strain AB-18-032, with corresponding sequences of representatives of the suborder *Micromonosporineae* showed that strain AB-18-032 is closely related to *Verrucosispora gifhornensis*, the sole representative of the genus *Verrucosispora* at this time (Figure 7).

The complete characterization of *Verrucosispora* sp. AB-18-032 was done by polyphasic taxonomy by the group of Michael Goodfellow (Newcastle University) [6]. Apparent from the combined phenotypic and genotypic data, the strain was classified in the genus *Verrucosispora* as a new species and described as *Verrucosispora maris* sp. nov. (=DSM 45365^T^ = NRRLB-24793^T^).

## 7. Increased Abyssomicin Spectrum

Several new members of the abyssomicin family were published during the last years. Abyssomicin E was isolated 2007 from terrestric *Streptomyces* sp. HKI0381. Abyssomicin E is similar in structure to abyssomicin D, and showed no biological activity [20]. In 2010, the structure of abyssomicin I produced by soil-derived *Streptomyces* sp. CHI39 was published. The compound exhibited inhibitory effects on tumor cell invasion [21]. One year later, the structures of ent-homoabyssomicins A and B were published, isolated from forest soil *Streptomyces* sp. Ank 210 [22]. Abyssomicins J, K, and L were isolated in 2013 from the South China Sea deep-sea sediment *Verrucosispora* sp. MS100128, showing anti-tuberculosis effects [23]. Abyssomicins 2–5 were produced by the marine-derived *Streptomyces* sp. RLUS1487 and were published in 2015. Abyssomicin 2 was identified as a selective reactivator of latent HIV virus [24]. In 2017, neoabyssomicins A–C were published, isolated from the deep-sea-derived *Streptomyces koyangensis* SCSIO. Neoabyssomicin A was found to augment HIV-1 virus replication in a human lymphocyte model [25]. In the same year, abyssomicins M–X were isolated from the coalmine fire strain *Streptomyces* sp. LC-6-2. The compounds were inactive in standard antimicrobial and cancer cell line cytotoxicity assays [26]. Abyssomicin Y was published in 2020 and is, to the best of my knowledge, the newest member of the abyssomicin family; it is produced by the marine-derived strain *Verrucosispora* sp. MS100137. The compound exhibited anti-influenza A virus activity [27].

The structures of the new members of the abyssomicin family are summarized in a review article by Sadaka et al. [9].

## Figures and Tables

**Figure 1 marinedrugs-19-00299-f001:**
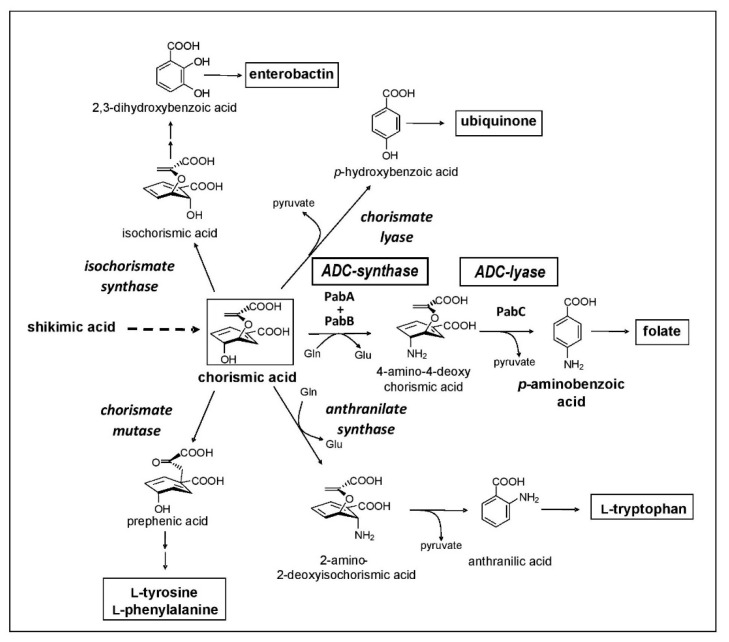
The bacterial chorismate biosynthetic pathway.

**Figure 2 marinedrugs-19-00299-f002:**
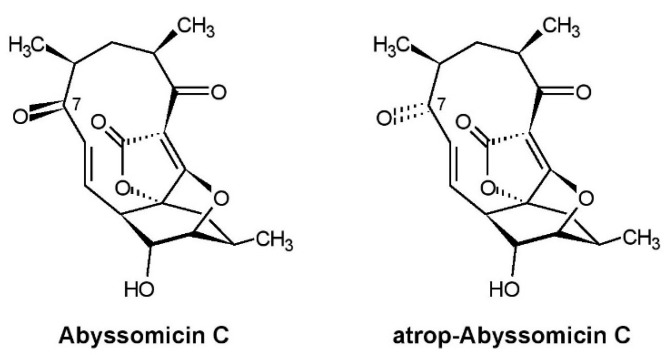
Structures of abyssomicin C and atrop-abyssomicin C produced by *Verrucosispora maris* AB-18-032.

**Figure 3 marinedrugs-19-00299-f003:**
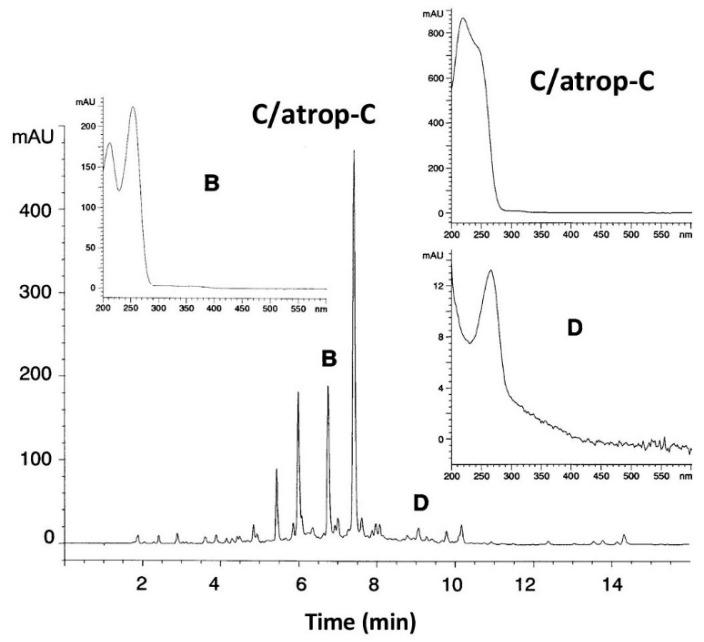
HPLC analysis of a culture filtrate extract from strain AB-18-032 at a fermentation time of 96 h, monitored at 260 nm. Inserts: UV-visible spectra of abyssomicin B, co-eluted C and atrop-C, and D. Reused with permission from [2], copyright owned by Japan Antibiotics Research Association.

**Figure 4 marinedrugs-19-00299-f004:**
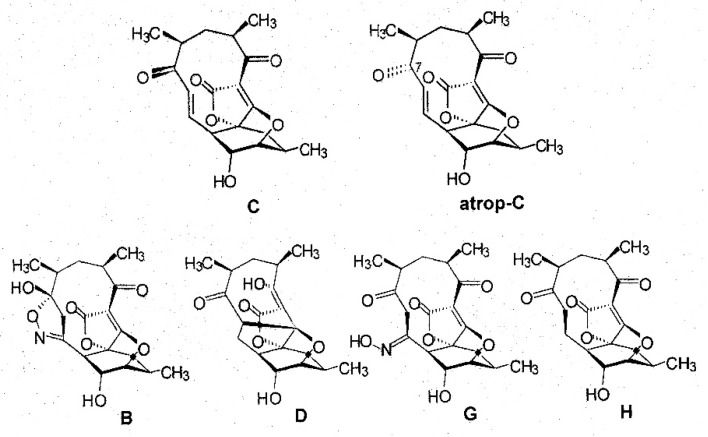
Structures of abyssomicins produced by *Verrucosispora maris* AB-18-032.

**Figure 5 marinedrugs-19-00299-f005:**
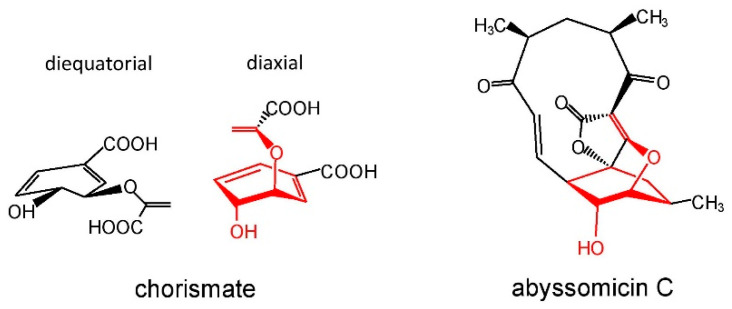
Similarity of the diaxial conformation of chorismate with the abyssomicin C core.

**Figure 6 marinedrugs-19-00299-f006:**
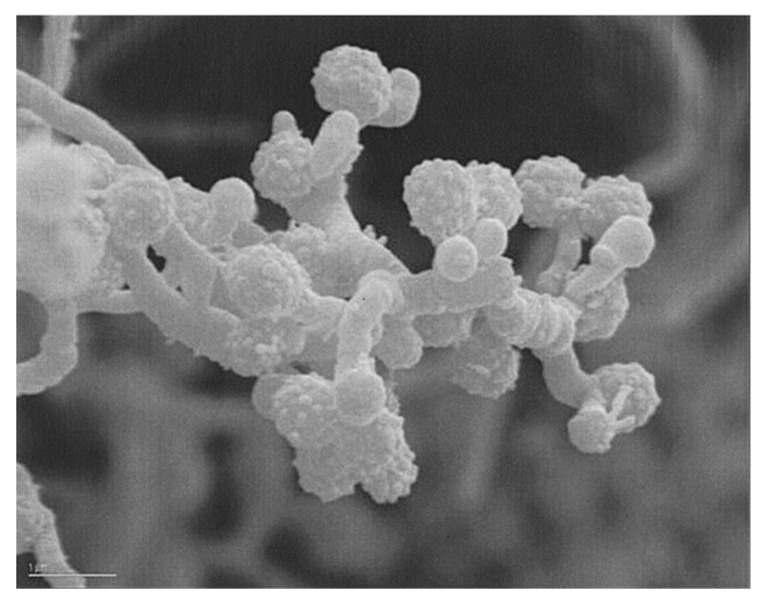
Scanning electron micrograph of *Verrucosispora* sp. AB-18-032. Bar: 1 µm. Reused with permission from [2], copyright owned by Japan Antibiotics Research Association.

**Figure 7 marinedrugs-19-00299-f007:**
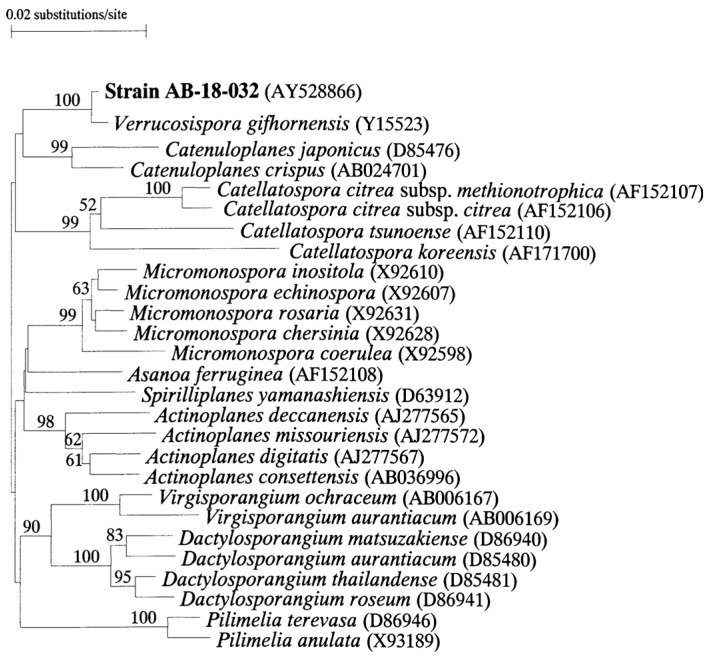
Neighbor-joining phylogenetic tree based on nearly complete 16S rDNA sequences of members of the suborder Micromonosporineae. Reused with permission from [2], copyright owned by Japan Antibiotics Research Association.

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
