# Peer review of "Abyssomicins—A 20-Year Retrospective View"

_marinedrugs, 2021, doi:10.3390/md19060299_

Round 1

Reviewer 1 Report

This article is an exciting review of abyssomicins' detection and isolation, and story, especially of abyssomicin C and its atropisomer. Nevertheless, the language is not very comprehensive and changes, such as using shorter sentences to explain things to make it more understandable are necessary. Also, punctuation (commas, full stops) and articles should be checked and added for better understanding.

Some suggestions include the following alterations:                

In the abstract section:

  1. Lines 11-14. The two sentences should be put in different order, to point out first the strong inhibitory activity of abyssomicin C and its atropisomer and then their mode of action. I think that this will make the text more concise. 
  2. Line 12. Add a comma before "which."
  3. Line 13. Replace "From all" with "Among the"
  4. Line 14. Replace "are showing" with "show." 

In the Introduction Section

  1. Line 22. Replace "the following" with "specific" or "a set of" 
  2. Line 22-25. You could make this sentence in two shorter as such: 

First, a massive set of…microorganisms, with a high potency of producing unique secondary metabolites. A representative example of such source are Actinomycetales, which are known…

  1. Line 25. Remove "which is" before "essential" and replace "which is" with "but" before "not"
  2. Line 26. Replace "toxic effects" with "selectivity in toxic effects."
  3. Line 29. Replace "by means of" with "utilizing" or "employing" 
  4. Line 32. Add "for" before "over"
  5. Lines 33-34. Remove "on the one hand" and :on the other hand"

Page 2 

  1. Line 36. Add a full stop after "biosynthesis" and start a new sentence with "This pathway is…" Also, add "such" before "as" 
  2. Line 37. Add "it" before "is" 
  3. Line 48 Remove "which was"
  4. Line 50. Put "selectively" at the end of the sentence
  5. Lines 53-56. This sentence makes no sense and should be written otherwise to point out its meaning
  6. Line 61. Replace "in the following" with "subsequently" and in general, put the adverbs at the end of a sentence and not within it

 Section 4. Pages 4-5

On page 4, I suggest the addition of "and Synthesis" in the title of section 4 since you report some synthetic strategies.

Lines 136-137. Move "besides abyssomicin C" to line 136, after "and"

Also, you could add some important features of structure elucidation, including interesting chemical properties, such as the interconversion of abyssomicin C to its atropisomer and vice versa. 

Page 5

Line 146. Add "the" before "topic", replace "synthesis" with "synthetic" 

You could also add some recent advances in synthesis to point out the continuous interest in abyssomicins and especially in the core of abyssomicin C. Such an example is European Journal of Organic Chemistry Volume 2020, Issue 29 p. 4547-4557, which is related to recent synthetic improvements in this area. Also, interesting reviews related to the isolation, mode of action, and synthesis of the abyssomicin family and abyssomicin C should also be included in the reference section since synthesis is analyzed in this paper briefly. Some examples are Lacoske and Theodorakis,  J. Nat. Prod. 201578, 562−575 and  Braddock and Theodorakis, Mar. Drugs 201917, 232−247. 

Page 5-6 

In section 5, you could add a scheme to show the similarity of the conformation of the abyssomicin C core with chorismate to show its mode of action more concisely.

Author Response

Answers to the reviewer suggestions:

Lines 11-14:          done

Line 12:                done

Line 13:                done

Line 14:                done

Line 22:                done

Lines 22-25:          done

Line 25:                done

Line 26:                done

Line 29:                done

Lind 32:                done

Lines 33-34:          done

Line 36:                done

Line 37:                done

Line 48:                done

Line 50:                done

Lines 53-56:          done

Line 61:                done

The title of Section 5 is changed to “Structure elucidation and synthesis strategies“

Lines 136-137:      done

Your recommendation “Add some important features of structure elucidation, including interesting chemical properties, such as the interconversion of abyssomicin C to its atropisomer and vice versa“ seems to be a problem for me, because detailed structure elucidation and interpretation of the results is absolutely not within my expertise. I am very sorry for this

Line 146:              done

Section 4, Paragraph 3: The recommended additional references were considered and included in the reference section.

Section 5: A new Figure (Figure 5) was added that shows the similarity of one solution conformation of chorismate with the abyssomicin C core.

I thank you very much for the very helpful comments to improve the manuscript.

Reviewer 2 Report

Very interesting paper on a neglected subject explaining both the manner of research and the results. Little is known about both microorganisms and their metabolites in marine environments.

Minor points:

- The weakest part is the biological activity of the compounds. Citing only MIC results for two strains is rather poor when time-kill curves give much more insights on the temporal action of killing the bacteria.

- The lines 46 to 50 and 107 to 110 mention nearly identical information.

- When collaborators are mentionned in the acknowledgments they should not be mentionned several times in the paper.

Author Response

Answers to the reviewer suggestions:

Section 5: Additional data on MIC values for significant multi-resistant and vancomycin-resistant pathogens are included, as “Gram-negative bacteria, filamentous fungi and yeasts were not sensitive against abyssomicin C and atrop-abyssomicin C“. At this time (2001), no time-kill curves were made.

Lines 46-50 and 107-110: The information is modified.

The Section “Acknowledgements“ is removed.

I thank you very much for the suggestions to improve the manuscript.

Reviewer 3 Report

The manuscript deals with a 20-years historic view of the development and analysis of Abyssomicins. The text is very easy to read and it describe all the processes required for the development of a new antibiotic. Also. the most recent members of the abyssomicin family are presented with the adequate number of references.

Good manuscript suitable for publication

Author Response

Thank you.